*Report*

EMBO
Molecular Medicine

# Sphingoid long chain bases prevent lung infection by *Pseudomonas aeruginosa*

Yael Pewzner-Jung[1],[**], Shaghayegh Tavakoli Tabazavareh[2], Heike Grassmé[2], Katrin Anne Becker[2], Lukasz Japtok[3], Jörg Steinmann[4], Tammar Joseph[1], Stephan Lang[5], Burkhard Tuemmler[6], Edward H Schuchman[7], Alex B Lentsch[8], Burkhard Kleuser[3], Michael J Edwards[8], Anthony H Futerman[1] & Erich Gulbins[2,8,*]

## Abstract

Cystic fibrosis patients and patients with chronic obstructive pulmonary disease, trauma, burn wound, or patients requiring ventilation are susceptible to severe pulmonary infection by *Pseudomonas aeruginosa*. Physiological innate defense mechanisms against this pathogen, and their alterations in lung diseases, are for the most part unknown. We now demonstrate a role for the sphingoid long chain base, sphingosine, in determining susceptibility to lung infection by *P. aeruginosa*. Tracheal and bronchial sphingosine levels were significantly reduced in tissues from cystic fibrosis patients and from cystic fibrosis mouse models due to reduced activity of acid ceramidase, which generates sphingosine from ceramide. Inhalation of mice with sphingosine, with a sphingosine analog, FTY720, or with acid ceramidase rescued susceptible mice from infection. Our data suggest that luminal sphingosine in tracheal and bronchial epithelial cells prevents pulmonary *P. aeruginosa* infection in normal individuals, paving the way for novel therapeutic paradigms based on inhalation of acid ceramidase or of sphingoid long chain bases in lung infection.

**Keywords** cystic fibrosis; long chain base; lung infection; *Pseudomonas aeruginosa*; sphingosine

**Subject Categories** Microbiology, Virology & Host Pathogen Interaction; Respiratory System

## Introduction

Cystic fibrosis (CF) is caused by mutations in the cystic fibrosis transmembrane conductance regulator (*CFTR*) gene and is found with a prevalence of 1 in 2,500–3,500 Caucasian newborns. CF patients, and patients with chronic obstructive pulmonary disease (COPD), are particularly susceptible to *Pseudomonas aeruginosa* infection, with approximately 80% of CF patients suffering from chronic *P. aeruginosa* pneumonia by 25 years of age, and one-third of patients with COPD hosting the bacteria (Rakhimova *et al*, 2009; Cystic Fibrosis Foundation, 2011). However, the increasing rate of *P. aeruginosa* resistance to many antibiotics necessitates the development of alternative therapies, which might also be beneficial in preventing *P. aeruginosa* infection in patients with trauma, burn wounds, sepsis, or in patients requiring ventilation (McManus *et al*, 1985; Crouch Brewer *et al*, 1996; Vidal *et al*, 1996).

Previous studies have demonstrated that various natural lipids and lyso-lipids act as bactericidal agents in skin (Fischer *et al*, 2012). Among these is the sphingoid long chain base (LCB), sphingosine (SPH), which protects human skin from bacterial colonization (Bibel *et al*, 1992). SPH is generated by hydrolysis of ceramide via acid ceramidase (AC). We now demonstrate that tracheal and bronchial epithelial SPH levels play a vital role in preventing normal individuals from *P. aeruginosa* lung infection. SPH levels are significantly reduced in tracheal and bronchial epithelia of CF patients and of CF mice, due to reduced AC activity, and normalization of SPH levels reverses susceptibility to *P. aeruginosa*. These data suggest two novel paradigms, namely that epithelial SPH is a natural antibactericidal agent in airways and that patients susceptible to bacterial infection in the lung could be treated by administration of AC or of SPH analogs.

1 Department of Biological Chemistry, Weizmann Institute of Science, Rehovot, Israel
2 Department of Molecular Biology, University Hospital Essen, University of Duisburg-Essen, Essen, Germany
3 Department of Nutritional Science, University of Potsdam, Potsdam, Germany
4 Department of Microbiology, University Hospital Essen, University of Duisburg-Essen, Essen, Germany
5 Department of Otorhinolaryngology, University Hospital Essen, University of Duisburg-Essen, Essen, Germany
6 Klinische Forschergruppe, OE 6710, Medizinische Hochschule Hannover, Hannover, Germany
7 Department of Genetics and Genomic Sciences, Icahn School of Medicine at Mount Sinai, New York, NY, USA
8 Department of Surgery, University of Cincinnati, Cincinnati, OH, USA
*Corresponding author. Tel: +49 201 723 3118; Fax: +49 201 723 5974; E-mail: erich.gulbins@uni-due.de
**Corresponding author. Tel: +972 8 9343256; Fax: +972 8 9344112; E-mail: yael.pewzner-jung@weizmann.ac.il

## Results and Discussion

Immunohistochemical analysis using an anti-SPH antibody demonstrated that SPH was abundantly expressed on the luminal surface of human nasal epithelial cells obtained from healthy individuals, but was almost undetectable on the surface of nasal epithelial cells from CF patients (Fig 1A – for validation of the specificity of the antibody, see Supplementary Information, Supplementary Fig S1 and Fig 2B). The reduction in SPH levels was recapitulated in tracheal and bronchial cells from CF mice (Fig 1B and C) and in airway epithelial cells from ceramide synthase 2 (*CerS2*)-null mice (Fig 1C). *CerS2*-null mice also displayed increased ceramide levels specifically in tracheal and bronchial epithelial cells (Fig 1D and Supplementary Fig S2), reminiscent of the elevated ceramide levels in these cells in CF mice and in CF patients (Fig 1D) (Teichgräber *et al*, 2008; Becker *et al*, 2010; Brodlie *et al*, 2010; Ulrich *et al*, 2010; Bodas *et al*, 2011). Thus, *CerS2*-null mice display similar changes in ceramide and SPH levels as in CF mice despite their different genetic alterations. AC inhalation increased surface SPH in bronchial epithelial cells of CF and *CerS2*-null mice (Fig 1C) with a concomitant reduction in ceramide levels (Fig 1D).

To quantify SPH levels in the lung, we established a number of innovative methods, including: (i) mass spectrometry (MS) and (ii) enzymatic assays for SPH and ceramide using extracts from freshly isolated tracheal epithelial cells, which detects total SPH levels in these cells, (iii) *in situ* enzymatic assays for SPH and ceramide using the respective kinases applied directly on intact tracheal surfaces, which detects SPH and ceramide exclusively on the luminal surface, and (iv) immunoprecipitation of SPH upon incubation of the anti-SPH antibody at the luminal surface of intact trachea, which also detects SPH exclusively on the luminal surface. First, freshly isolated tracheal epithelial cells were extracted and SPH assayed using MS and enzymatic assays for SPH, demonstrating an approximately 75% reduction in total SPH levels in CF mice (Fig 2A). Next, an *in situ* enzyme assay, performed by application of SPH kinase (SK) and [$^{32}$P]γATP directly to the luminal side of the intact tracheal epithelial cell layer, revealed an approximately 75% reduction in SPH levels (Fig 2B). The reduced SPH on the tracheal surface was confirmed by SPH immunoprecipitation using the anti-SPH antibody coupled to protein L-agarose beads, followed by lipid extraction and an enzymatic assay for SPH (Fig 2B). Application of AC to the surface of isolated CF trachea prior to the *in situ* enzyme assay normalized SPH levels (Fig 2B). Incubation of the isolated tracheal surface with 10 μM cytochalasin B (an actin filament polymerization inhibitor) prevented *P. aeruginosa* internalization into tracheal epithelial cells, but did not alter the amount of SPH detected by the *in situ* enzyme assay for SK or by SPH immunoprecipitation, excluding the possibility that SK and/or antibody internalization occurs during the assay (Fig 2B). These results

demonstrate that SPH is present on the surface of WT epithelial cells while almost completely absent on the surface of CF epithelia.

We next demonstrated that AC or SPH inhalation increased SPH levels in CF tracheal epithelial cells and on the surface of CF trachea *in vivo* (Fig 2A and B).

Moreover, significant accumulation of ceramide was detected by mass spectrometry (MS) (Fig 2C, left) in extracts of isolated CF epithelial cells and by *in situ* kinase assay on the luminal surface of these cells in trachea of CF mice (Fig 2C, right), which was corrected by inhalation of AC (Fig 2C). The specificity of the enzyme assay was confirmed by treating isolated trachea with AC *in vitro* (Fig 2C, right).

To determine the mechanism by which SPH levels are decreased on the surface of CF tracheal epithelial cells, AC activity was analyzed by loading trachea with [$^{14}$C]C16-ceramide and its consumption *in vivo* was analyzed. Significantly lower levels of AC activity were detected in CF mice (Fig 2D). *Cftr* regulates the pH in secretory lysosomes (Barasch *et al*, 1991; Teichgräber *et al*, 2008) and probably also regulates the pH in small domains on the surface of tracheal epithelial cells that express proton pumps (Xu *et al*, 2012) by providing Cl$^-$ counterions for H$^+$, thus permitting continuous activity of protons pumps. Moreover, since alkalization results in a marked inhibition of AC activity (Li *et al*, 1998; Teichgräber *et al*, 2008), we next demonstrated that surface acidification of the trachea restored AC activity, increased surface sphingosine and decreased ceramide levels (Fig 2D). To test whether acid-mediated hydrolysis of ceramide contributes to its consumption and the generation of sphingosine, we performed the experiment in the presence of the AC inhibitors, carmofur, or oleoylethanolamine. The inhibitors completely prevented sphingosine generation excluding significant acid-mediated hydrolysis of ceramide (Fig 2D). It is important to note that the pH of small microdomains on the cell membrane of CF cells is independent of the airway liquid surface pH, which is decreased from pH 7.2 to pH 6.8 in CF cells (Pezzulo *et al*, 2012), which is probably not enough to alter AC activity.

Next, we examined whether sphingosine plays a role in infections with *P. aeruginosa*. We used three different *P. aeruginosa* strains to exclude any strain-specific effects. Upon intranasal infection, CF and *CerS2*-null mice displayed a dramatically increased sensitivity to *P. aeruginosa* strains 762 (Fig 3A), PA14, and ATCC 27853 (Fig 3B), with 10- to 100-fold more bacteria in the lung 3–4 h after infection (Fig 3A and B), clinical signs of severe pneumonia (Supplementary Fig S3A and B), massive release of cytokines such as interleukin-1β (IL-1β) (Fig 3C) and TNF-α (Supplementary Fig S4) and influx of leukocytes (Supplementary Fig S5A–H). Inhalation of CF mice with SPH to a level similar to that found in WT mice (Fig 2B), of AC or of FTY720 (a SPH analog in clinical use for treating multiple sclerosis) 1 h prior to *P. aeruginosa* infection, protected CF and *CerS2*-null mice from infection (Fig 3A–C and

**Figure 1.  Immunohistochemical analysis of SPH.**

A–D    Immunostaining of paraffin sections of nasal tissues from (A) healthy and CF individuals or (B) trachea and (C) bronchi of WT, CF, and *CerS2*-null mice using a Cy3-coupled anti-SPH antibody. Effect of AC inhalation on SPH (C) and ceramide levels (D) in CF and *CerS2*-null mice. Representative images are shown; fluorescence levels were quantified and are given in arbitrary units (a.u.) (means ± s.d., *n* = 5 in A, *n* = 4 in B, *n* = 6 for WT controls for CF mice, *n* = 8 for CF, and *n* = 4 for all others in C, and *n* = 9 for untreated WT or CF, *n* = 7 for AC-inhaled WT, *n* = 8 for AC-inhaled CF and *n* = 4 for *CerS2*-null mice in D). Numbers above bars indicate the exact calculated *P*-values.

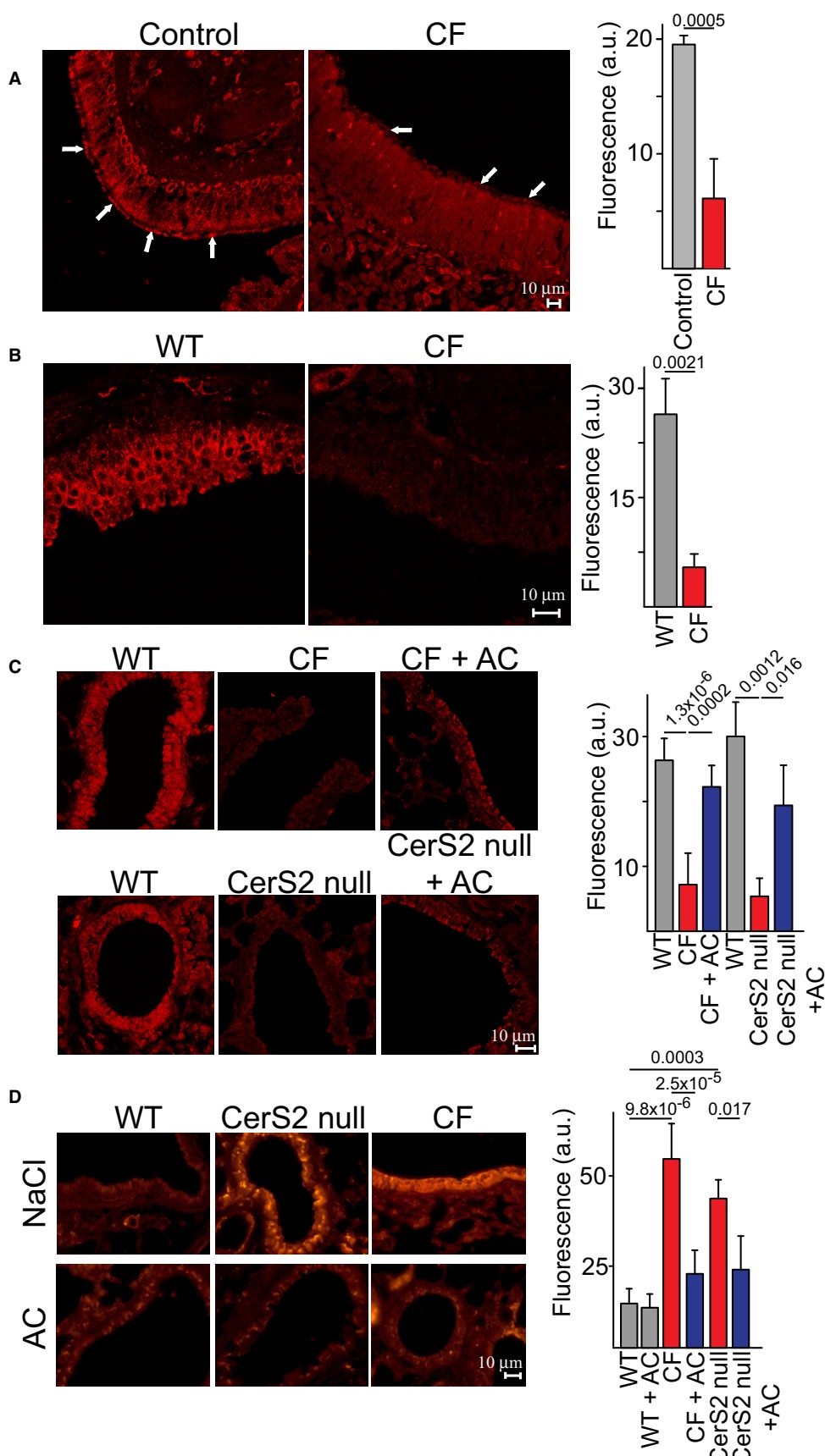

**Figure 1.**

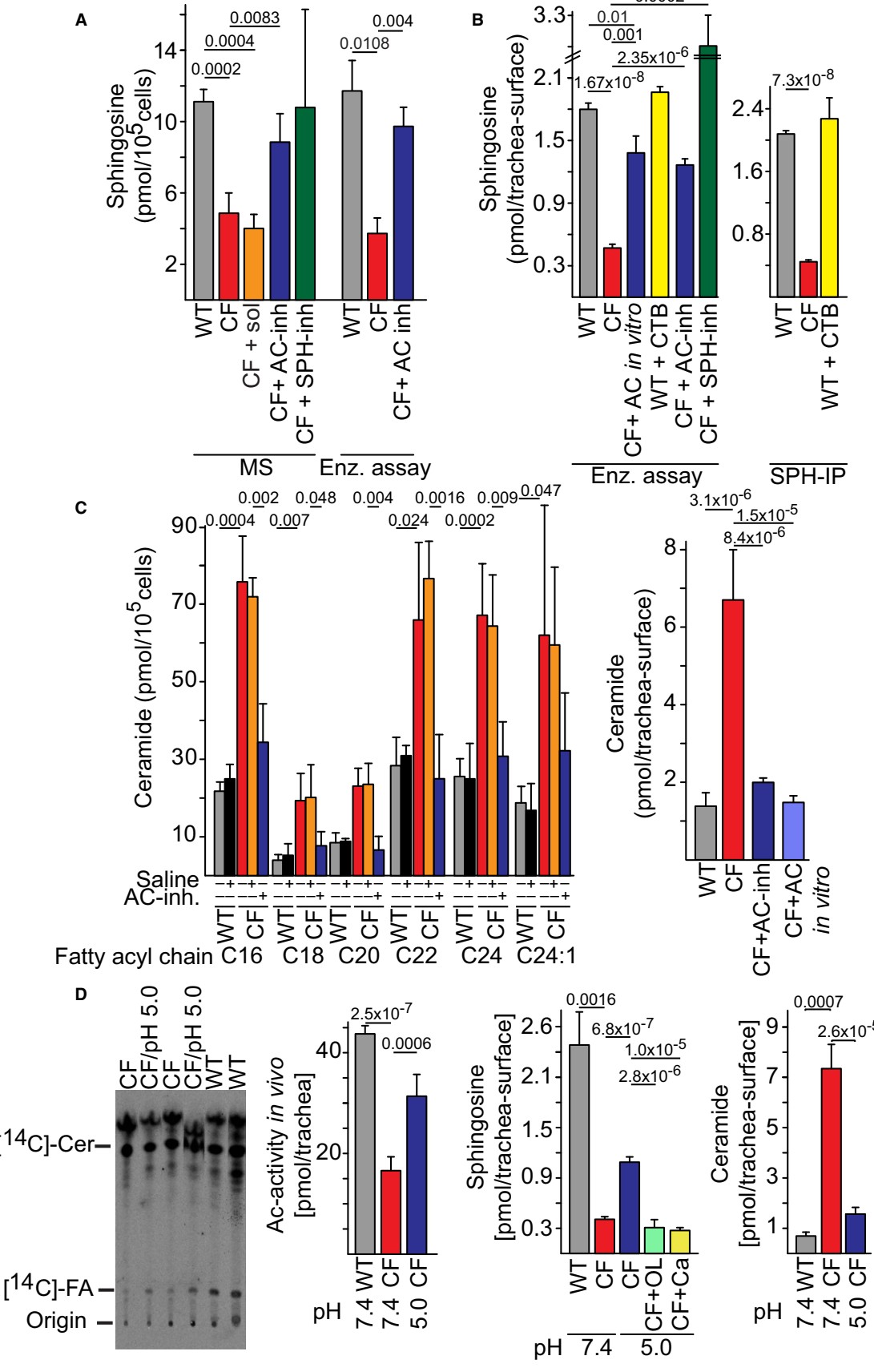

**Figure 2.**

Supplementary Figs S3, S4 and S5). To test whether sphingosine and FTY720 are also able to cure an existing pulmonary infection, mice were inhaled with SPH or FTY720 1 h after infection, a stage at which the mice show the first signs of infection, with both completely blocking the development of infection (Fig 3A–C, Supplementary Figs S3, S4 and S5). The low bacterial numbers and the lack of cytokines and leukocyte influx demonstrate that sphingosine and FTY720 directly and rapidly kill the bacteria, thus preventing further inflammation.

Controls on non-infected mice showed that inhalation of SPH, AC, or FTY720 had no influence on pulmonary cytokines nor did it induce neutrophil influx 4 h, 1, 7, or 14 days after inhalation (Fig 3C, Supplementary Figs S4, S5A, B, E, F, and S6A–C).

*P. aeruginosa* was next incubated *in vitro* with a number of natural and synthetic sphingoid LCBs. A number of LCBs inhibited bacterial growth with $EC_{50}$ values of 0.3–2.2 μM, including the plant LCB, phytosphingosine, non-natural stereoisomers of SPH, and dihydrosphingosine (sphinganine), as well as LCBs which have recently been discovered to occur at low levels in mammals (Pruett *et al*, 2008) (Table 1). A number of other LCBs also inhibited bacterial growth with a somewhat higher $EC_{50}$ value, while other LCBs were without effect at concentrations as high as 50 μM (Table 1). These results demonstrate a notable specificity of the effect of LCBs on bacterial growth, providing structural information that could be used in development of novel LCB-based drugs to treat lung infection. To investigate whether other bacteria that affect CF patients are also susceptible to LCB treatment, we examined the effect of sphingosine on *Acinetobacter baumannii*, *Haemophilus influenzae*, *Burkholderia cepacia*, and *Moraxella catarrhalis*. Sphingosine inhibited growth of and/or killed *Acinetobacter baumannii* with an $EC_{50}$ of 0.07 ± 0.05 μM, of *Moraxella catarrhalis* with an $EC_{50}$ of 0.04 ± 0.004 μM of *Haemophilus influenzae* with an $EC_{50}$ of 4.8 ± 0.49 μM and *Burkholderia cepacia* with an $EC_{50}$ of 45 ± 6.3 μM. The relative resistance of *Burkholderia cepacia* might lie in its outer membrane unique composition (Cox & Wilkinson, 1991).

The results presented herein uncover a major innate defense mechanism of healthy airways, namely the bactericidal effect of lung and tracheal epithelial SPH. Upon loss of surface SPH, individuals become susceptible to *P. aeruginosa* infection, as exemplified in CF patients and mice, and in *CerS2*-null mice. Moreover, restoration of surface SPH by inhalation of SPH, of other LCBs or with AC, reverses susceptibility and cures existing *P. aeruginosa* infection,

suggesting that LCB inhalation might provide a novel therapeutic option to counteract pulmonary infection by bacteria. This is of great importance since many *P. aeruginosa* strains are multi-resistant to antibiotics, rendering pulmonary infections difficult to treat. Acidification of the trachea, which increases AC activity, leads to a reduction in ceramide levels and an increase in sphingosine levels, consistent with the clinical use of hypertonic saline (pH 5.0–5.5) in treating CF patients (Elkins *et al*, 2006).

Ours is not the first study to identify SPH as an anti-bacterial agent (Bibel *et al*, 1992; Arikawa *et al*, 2002; Fischer *et al*, 2013). However, it is the first to show that LCBs are efficacious in acute lung infection. The mechanism of bacterial cell death may involve a direct effect of the LCBs on the bacteria (Fischer *et al*, 2012), or possibly up-regulation of porin-like proteins that are responsible for channel formation in the bacterial membrane (LaBauve & Wargo, 2014). Information obtained by comparing LCB structures suggests that longer and more positively charged LCBs are more effective than shorter, negatively charged LCBs; moreover, modification of the C-1 position of the LCB has variable effects on bacterial survival. Based on this structural information, we predict that it will be possible to generate LCBs that are highly efficacious in killing bacteria without causing effects on the host. If this notion is correct, then these putative novel LCBs analogs could pave the way for generation of drugs used for inhalation against *P. aeruginosa* infection and possibly infection by other bacteria.

## Materials and Methods

### Mice

*CerS2*-null and CF mice were generated as described (Charizopoulou *et al*, 2006; Pewzner-Jung *et al*, 2010). Two different CF mouse strains were used, $Cftr^{tm1Unc}$-$Tg^{(FABPCFTR)}$, abbreviated $Cftr^{KO}$ (mice that lack *Cftr* but express human CFTR in the gut under the control of a fatty acid binding protein promoter), obtained from Jackson Laboratories (Bar Harbor, ME, USA) and backcrossed for 10 generations onto a C57BL/6 background. All data shown in the current paper are from $Cftr^{KO}$ mice. To confirm data obtained with this strain, we used an additional strain, that is B6.129P2(CF/3)-$Cftr^{TgH}$ $^{(neoim)Hgu}$ (abbreviated $Cftr^{MHH}$) congenic mice that were established by brother–sister mating from the original $Cftr^{TgH(neoim)Hgu}$ mutant

---

◄

**Figure 2.  Biochemical analysis of SPH and ceramide levels.**

A, B   (A) SPH levels were determined by mass spectrometry (MS) or an enzymatic kinase assay (Enz. assay) in lysates from isolated tracheal epithelial cells or (B) by an *in situ* enzymatic kinase assay on the tracheal surface (left) or by immunoprecipitation of SPH (SPH-IP) from the luminal surface followed by quantification using an enzymatic assay (right). Pre-incubation of WT trachea with cytochalasin B (CTB) did not change SPH surface levels. Incubation of trachea *in vitro* with AC proved the specificity of the enzymatic assay. Inhalation (inh) of AC (200 units) or SPH normalized total SPH levels in isolated tracheal epithelial cells (A) and on the luminal surface (B), the solvent (sol) was without effect.

C   Ceramide species in isolated tracheal epithelial cells were measured by MS (left). Ceramide on the luminal surface was determined by an *in situ* enzymatic kinase assay (right). AC inhalation corrected increased ceramide levels in CF mice, incubation of the luminal surface with AC *in vitro* served to prove the specificity of the *in situ* kinase assay (right).

D   $[^{14}C]C16$-ceramide ($[^{14}C]$-Cer) was injected into the trachea of anesthesized mice and AC activity determined. Acidification was achieved by injection of $[^{14}C]C16$-ceramide in 150 mM sodium acetate, pH 5.0. Sphingosine and ceramide levels were determined at two different pHs in isolated tracheae from CF mice. Tracheae were incubated in 150 mM sodium acetate pH 5.0 or pH 7.4 for 30 min prior to analysis. Tracheae were also treated with the AC inhibitors oleoylethanolamine or carmofur to exclude acid-mediated hydrolysis of ceramide.

Data information: Data are means ± s.d., *n* = 4. Numbers above bars indicate the exact calculated *P*-values.

---

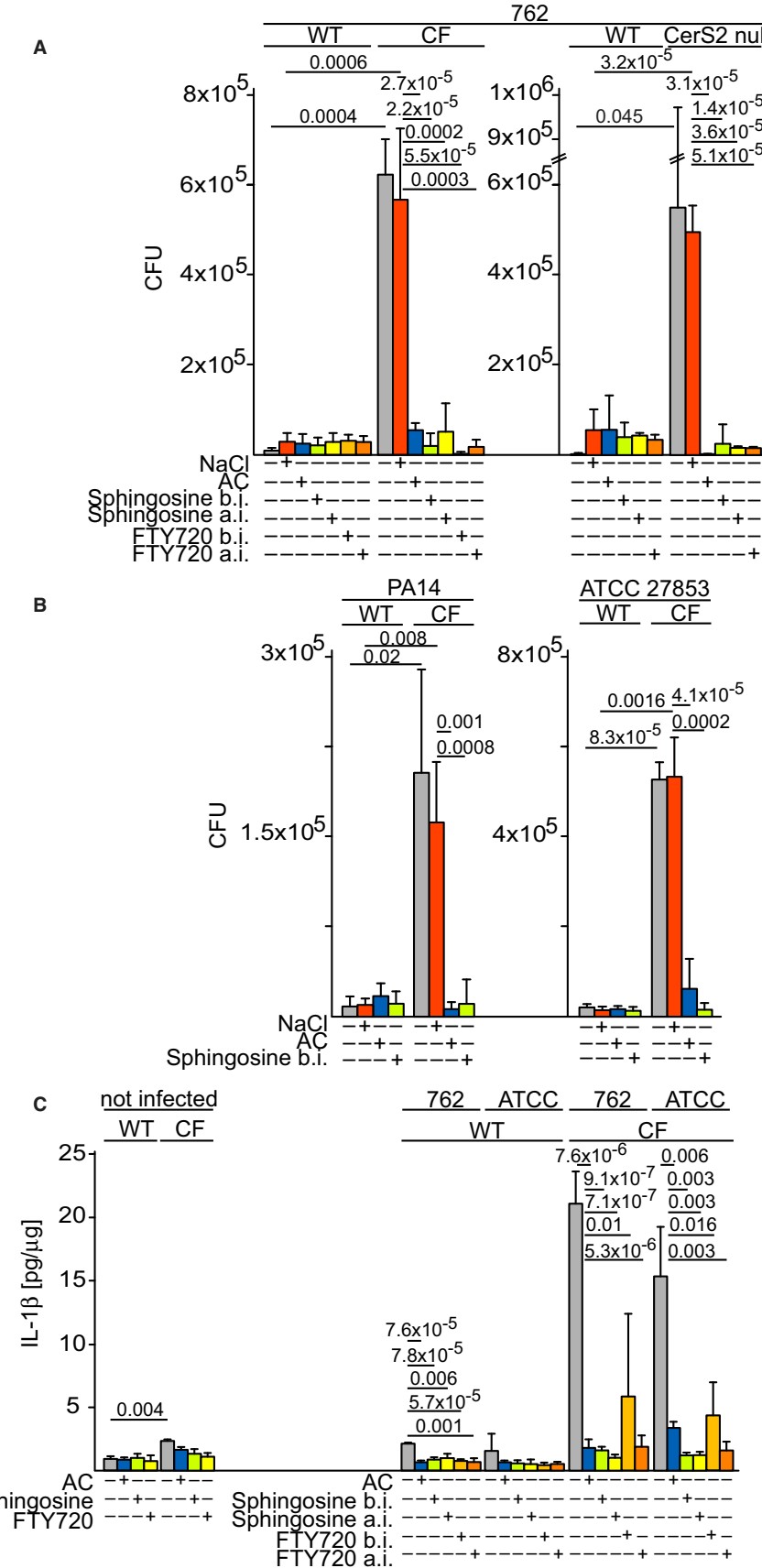

**Figure 3.**

mouse that was generated using insertional mutagenesis in *Cftr* exon 10. Congenic *Cftr^MHH* mice were generated by backcrossing the targeted mutation onto the B6 inbred background. The strain produces low levels of Cftr. *CerS2*-null mice were on an F1 background achieved by intercross of heterozygous *CerS2^{GT/+}*-C57BL/6 mice (GT = gene trap) and heterozygous *CerS2^{GT/+}*-129S4/SvJae mice, to generate *CerS2^{GT/GT}* F1, and CerS2^{+/+} F1 mice used in the experiments as *CerS2*-null and wild-type control, respectively.

Control mice were syngenic littermates (C57BL/6 for CF mice and C57BL/6;129S4/SvJae WT for *CerS2*-null mice). The cystic fibrosis and the corresponding wild-type mice were used at an age of 16–18 weeks. We used female CF and WT mice in the present study. The CerS2-deficient mice were both male and females and used at an age of 6–8 weeks. Mice were bred in a special pathogen free (SPF) facility at the Weizmann Institute of Science and in the University of Duisburg-Essen. Mice were handled according to protocols approved by the Weizmann Institute of Science and the University of Duisburg-Essen Animal Care Committee as per international guidelines.

### Immunohistochemical analysis of ceramide and SPH and light microscopy

Animals were sacrificed, lungs or trachea immediately removed, fixed for 36 h in 4% paraformaldehyde (PFA) in phosphate buffer saline (PBS), embedded in paraffin, and trimmed to 6 μm. Sections were dewaxed, incubated for 30 min with Pepsin Digest All (Invitrogen Life Technologies, USA) at 37°C, washed, and blocked for 10 min with PBS, 0.05% Tween-20, and 5% fetal calf serum (FCS). Sections were subsequently immunostained for 45 min with an anti-ceramide antibody (clone S58-9, Glycobiotech, Germany), an anti-SPH antibody (clone NHSPH, Alfresa Pharma Corporation, Japan), or an anti-Gr1-antibodies (BD, Germany). Anti-ceramide antibodies were diluted 1:100, anti-SPH antibodies 1:1,000, anti-Gr1 antibodies 1:200 in H/S (132 mM NaCl, 20 mM HEPES, pH 7.4, 5 mM KCl, 1 mM CaCl$_2$, 0.7 mM MgCl$_2$, 0.8 mM MgSO$_4$) plus 1% FCS, washed three times in PBS plus 0.05% Tween-20, incubated for 45 min with Cy3-coupled donkey anti-mouse IgM F(ab)$_2$ fragments or Cy3-coupled donkey anti-rat IgG (Jackson ImmunoResearch), washed again three times in PBS containing 0.05% Tween-20 and once in PBS, and embedded in coverslip mounting solution (Mowiol). Immunostained samples were analyzed on a confocal Leica TCS SL or SP5 microscope (Leica, Germany). Immunofluorescence was analyzed using Leica software, version 2.61, in 10–25 areas per sample, by measuring the fluorescence in the apical one-third of the bronchial or tracheal epithelial cells.

To exclude binding of the anti-SPH antibody to sphinganine, the trachea from WT or CF mice were removed, incubated *in vitro* with 10 μM sphinganine in 150 mM sodium acetate (pH 7.4) for 30 min, washed three times in 150 mM sodium acetate (pH 7.4), fixed for 36 h in 2% PFA (pH 7.3), and stained with the anti-SPH antibody. Hemalaun was used to stain lung sections for light microscopy.

### Mouse inhalation

Inhalation was performed using a PARI Boy SX nebulizer (PARI GmbH, Starnberg, Germany), which generates a fine aerosol by pumping the fluid with an air jet. Mice were inhaled with the aerosol via a mask that is part of an oral inhalation device for children (LL-Nebulizer); the mask was clipped at the sides to cover only the nose and the surrounding part of the face. Mice were inhaled with 800 μl of 0.9% NaCl containing SPH (125 μM), AC (80 μg of purified protein), or FTY720 (125 μM). Approximately 10% of the volume that is applied to the mice is inhaled.

### Enzymatic assays to measure ceramide and SPH analysis *in situ*

Mice were sacrificed and the trachea immediately removed. The trachea was carefully opened, washed in 150 mM sodium acetate (pH 7.4), placed on a 30°C pre-warmed plastic plate, and incubated with 0.01 units diacylglycerol (DAG) kinase (Biomol, Germany) to measure ceramide, or with 0.001 units SK (R&D Systems, Germany) to measure SPH, in 4 μl of 150 mM sodium acetate (titrated to pH 7.4), 1 mM adenosine triphosphate (ATP), and 10 μCi [$^{32}$P]γATP (for the buffer composition please see also Supplementary Fig S7). Controls were incubated with the same buffer without SK or left untreated. We ensured that the kinase buffer was only added to the luminal surface of the trachea. The kinase reaction for ceramide was performed for 15 min at 30°C and terminated by transfer of the trachea into CHCl$_3$:CH$_3$OH:1N HCl (100:100:1, v/v/v) followed by addition of 170 μl buffered saline solution (135 mM NaCl, 1.5 mM CaCl$_2$, 0.5 mM MgCl$_2$, 5.6 mM glucose, 10 mM HEPES, pH 7.2) and 30 μl of 100 mM EDTA. The SK reaction was terminated by placing the trachea in 100 μl H$_2$O, followed by addition of 20 μl 1N HCl, 800 μl CHCl$_3$/CH$_3$OH/1N HCl (100:200:1, v/v/v), 240 μl CHCl$_3$, and 2 M KCl. The lower phase was collected, dried, dissolved in 20 μl of CHCl$_3$:CH$_3$OH (1:1, v/v), and separated on Silica G60 thin layer chromatography (TLC) plates using CHCl$_3$/acetone/CH$_3$OH/acetic acid/H$_2$O (50:20:15:10:5, v/v/v/v/v) as developing solvent for ceramide and CHCl$_3$/CH$_3$OH/acetic acid/H$_2$O (90:90:15:5, v/v/v/v) for SPH. The TLC plates were exposed to radiography films, spots were removed from the plates, and the incorporation of [$^{32}$P] into ceramide measured by liquid scintillation counting. Ceramide and SPH were determined using a standard curve of C16-ceramides to C24-ceramides or C18-SPH. Surface pH was varied by adjusting the pH of the 150 mM sodium acetate buffer from 7.4 to 5.0 and incubation of the trachea for 20 min prior to the *in situ* kinase assay. To exclude acid-mediated lysis of ceramide, we pre-incubated the trachea for 15 min with 0.5 mM oleoylethanolamine (Calbiochem) or 1 μM carmofur (Sigma) prior to adjusting the pH.

◀ **Figure 3.    Effect of inhalation of AC and LCBs.**

A–C    Number of bacteria in the lungs (A and B) and the concentration of interleukin-1β (IL-1β) (C) after intranasal infection with $1 \times 10^8$ CFU of *P. aeruginosa* strains 762 (A, C), or of CF mice with strains PA14 and ATCC 27853 (B, C). Mice were inhaled with 0.9% NaCl or AC 1 h prior to infection (before infection, b.i.), or with SPH or FTY720 1 h before or 1 h after infection (a.i.). Data are means ± s.d., *n* = 4 for (A) and (B), *n* = 3 for (C). Numbers above bars indicate the exact calculated *P*-values.

**Table 1.  Effect of sphingoid LCBs on *P. aeruginosa* growth *in vitro*.**
$EC_{50}$ values were determined from at least three independent experiments. Values are means $\pm$ s.d.

| Long chain base | $EC_{50}$ ($\mu$M) |
|---|---|
| Inhibitors | |
| D-*erythro*-C20-sphingosine | 0.3 $\pm$ 0.1 |
| L-*threo*-C18-dihydrosphingosine (Safingol) | 0.4 $\pm$ 0.1 |
| D-*erythro*-C18-sphingosine | 0.5 $\pm$ 0.1 |
| 3-*deoxy*-D-*erythro*-C18-sphingosine | 0.5 $\pm$ 0.0 |
| D-*threo*-C18-dihydrosphingosine | 0.5 $\pm$ 0.1 |
| D-*threo*-C18-sphingosine | 0.6 $\pm$ 0.1 |
| D-*erythro*-C18-dihydrosphingosine | 0.6 $\pm$ 0.5 |
| L-*threo*-C18-sphingosine | 0.6 $\pm$ 0.2 |
| 1-*deoxy*-D-*erythro*-C18-dihydrosphingosine | 0.7 $\pm$ 0.0 |
| 1-desoxymethyl-C18-sphingosine | 0.7 $\pm$ 0.1 |
| L-*erythro*-C18-dihydrosphingosine | 0.8 $\pm$ 0.1 |
| L-*erythro*-C18-sphingosine | 0.8 $\pm$ 0.1 |
| 1-*deoxy*-D-*erythro*-C18-sphingosine | 0.9 $\pm$ 0.1 |
| Phytosphingosine | 0.9 $\pm$ 0.4 |
| D-*erythro*-C18-sphingosine | 1.0 $\pm$ 0.2 |
| FTY720 | 1.9 $\pm$ 0.4 |
| Monomethyl D-*erythro*-C18-sphingosine | 1.9 $\pm$ 0.1 |
| Galactosyl D-*erythro*-C18-sphingosine (psychosine) | 2.2 $\pm$ 1.4 |
| Non-inhibitors | |
| D-*erythro*-C12-sphingosine | > 50 |
| D-*erythro*-C14-sphingosine | > 50 |
| N-acetyl-D-*erythro*-C18-dihydrosphingosine | > 50 |
| Fumonisin B1 | > 50 |
| Sphingosine-1-phosphate | > 50 |

## SPH analysis in extracts of freshly isolated tracheal epithelial cells

Epithelial cells were removed from the trachea by carefully scraping the inner surface of the trachea. Cells were extracted in $CHCl_3$/$CH_3OH$/1N HCl (100:200:1, v/v/v), and the lower phase was dried and resuspended in a detergent solution (7.5% [w/v] n-octyl glucopyranoside, 5 mM cardiolipin in 1 mM diethylenetriaminepentaacetic acid [DTPA]). The kinase reaction was initiated by addition of 0.001 units SK in 50 mM HEPES (pH 7.4), 250 mM NaCl, 30 mM $MgCl_2$ 1 mM ATP, and 10 $\mu$Ci [$^{32}$P]$\gamma$ATP. Samples were incubated for 30 min at 37°C with shaking (350 rpm) and processed as above.

## Mass spectrometry

Ceramides and SPH in isolated tracheal epithelial cells were extracted and quantified as described *(Fayyaz et al, 2014)*. Sample analysis was carried out by rapid-resolution liquid chromatography-MS/MS using a Q-TOF 6530 mass spectrometer (Agilent Technologies, Waldbronn, Germany) operating in the positive ESI mode. The precursor ions of SPH (m/z 300.289), C17-SPH (m/z 286.274), and

ceramides (C16-ceramide (m/z 520.508), C17-ceramide (m/z 534.524), C18-ceramide (m/z 548.540), C18:1-ceramide (m/z 546.524), C20-ceramide (m/z 576.571), C22-ceramide (m/z 604.602), C24-ceramide (m/z 632.634), C24:1-ceramide (m/z 630.618)) were cleaved into the fragment ions of m/z 282.280, m/z 268.264, and m/z 264.270, respectively. Quantification was performed with Mass Hunter Software (Agilent Technologies).

## Enzyme assays of SPH immunoprecipitates

The luminal epithelial surface of the trachea was incubated with 50 ng of the anti-SPH antibody, that was pre-coupled to protein L-agarose beads (Santa Cruz Inc., Heidelberg, Germany), at 4°C for 30 min in 150 mM sodium acetate (pH 7.4) plus 1% FCS. Alternatively, the antibody was bound to the surface of the trachea by incubation at 4°C for 20 min in 150 mM sodium acetate (pH 7.4) plus 1% FCS. Trachea was washed three times in 150 mM sodium acetate (pH 7.4), and L-agarose was added. Samples were incubated for an additional 20 min, washed six times with ice-cold H/S buffer, and lysed in 125 mM NaCl, 25 mM Tris–HCl (pH 7.4), 10 mM EDTA, 10 mM sodium pyrophosphate, 3% Nonidet P-40, and 10 $\mu$g/ml aprotinin and leupeptin for 10 min on ice. Insoluble tissue was removed, and the agarose beads washed five times in lysis buffer. The beads were pelleted, the supernatants discarded, the beads were resuspended in 200 $\mu$l $H_2O$ and extracted using 600 $\mu$l $CHCl_3$/$CH_3OH$/1N HCl (100:200:1, v/v/v). The lower organic phase was collected, dried, resuspended in 20 $\mu$l of a detergent solution (7.5% [w/v] n-octyl glucopyranoside, 5 mM cardiolipin in 1 mM diethylenetriaminepentaacetic acid [DTPA]), and sonicated for 10 min. The kinase reaction was started by addition of 70 $\mu$l of a reaction mixture containing 2 $\mu$l SK in 50 mM HEPES (pH 7.4), 250 mM NaCl, 30 mM $MgCl_2$ 1 mM ATP, and 10 $\mu$Ci [$^{32}$P]$\gamma$ATP. The kinase reaction was terminated after 30 min, and samples extracted and processed as above.

## Cytochalasin B treatment

Trachea was incubated with 10 $\mu$M cytochalasin B for 20 min prior to the kinase assay or prior to SPH immunoprecipitation experiments. Surface SPH was measured by the *in situ* kinase assay or by immunoprecipitation as above. As controls, trachea was incubated with cytochalasin B prior to infection with $1 \times 10^6$ CFU *P. aeruginosa* ATCC 27853. The trachea was washed after infection, incubated for 60 min with 100 $\mu$g/ml polymixin, washed again, lysed in 5 mg/ml saponin for 10 min, and centrifuged (10 min, 1,600 *g*). Pellets were resuspended in H/S buffer, aliquots were plated on TSA, and grown overnight prior to quantification of the number of intracellular bacteria.

## AC activity assay

Mice were anesthetized with ketamine and xylazine and 4 $\mu$l [$^{14}$C]C16-ceramide (ARC0831, 55 mCi/mmol) injected into the lumen of the trachea. Prior to injection, [$^{14}$C]C16-ceramide was dried, resuspended in 0.5% OGP in 150 mM sodium acetate (pH 7.4 or 5.0), and bath sonicated for 10 min. After 20–30 min, mice were sacrificed, the trachea was carefully removed and extracted in 200 $\mu$l $H_2O$ and $CHCl_3$:$CH_3OH$:HCl (100:100:1, v/v/v). The lower phase

was dried, samples were resuspended in $CHCl_3:CH_3OH$ (1:1, v/v) and separated by TLC using $CHCl_3:CH_3OH:ammoniumhydroxide$ (90:20:0.5, v/v/v) as the developing solvent. The TLC plates were analyzed using a Fuji-Imager.

**Bacterial infection**

*P. aeruginosa* strains 762, PA14, and ATCC 27853 were grown for 14–14.5 h on fresh tryptic soy agar (TSA) plates (Becton Dickinson Biosciences, Germany). Bacteria were then transferred into 40 ml of pre-warmed, sterile tryptic soy broth (TSB) (Becton Dickinson Biosciences) in Erlenmeyer flasks. The OD was adjusted to 0.225, and the bacteria were grown for 60 min at 37°C with shaking at 125 rpm to obtain bacteria in the early log phase. Bacteria were centrifuged (10 min, 1,600 $g$), the supernatant removed, and bacteria resuspended in pre-warmed H/S buffer (132 mM NaCl, 20 mM HEPES, pH 7.4, 5 mM KCl, 1 mM $CaCl_2$, 0.7 mM $MgCl_2$, 0.8 mM $MgSO_4$). The OD was determined, and the bacteria diluted in pre-warmed H/S to a concentration of $1 \times 10^8$ CFU per 20 µl. This dose was previously shown to induce a severe infection in CF-mice, but only a mild infection in C57BL/6 mice (Teichgräber *et al*, 2008). Mice were anesthetized for 10–15 s, and bacteria carefully injected into the nose with a 30-gauge 1-ml syringe. The needle was covered with a tightly fitting, smooth plastic tube so that nasal injuries were avoided. Lungs were subsequently removed and homogenized and lysed for 10 min in 5 mg/ml saponin to release intracellular bacteria. Samples were centrifuged (10 min, 1,600 $g$), washed once with sterile H/S solution, and homogenates cultured on TSA plates. The number of bacteria (colony forming units, CFU) was analyzed after 18 h.

**Incubation of bacteria with sphingolipids**

The clinical *P. aeruginosa* strain 762 or clinical strains of *Acinetobacter baumanii*, *Moraxella catarrhalis*, and *Burkholderia cepacia* were grown in LB medium at 37°C until the early logarithmic phase ($OD_{550}$ of 0.2). Bacteria were diluted in PBS (without $Ca^{2+}$ and $Mg^{2+}$) to a concentration of $5 \times 10^3$ CFU/ml. *Haemophilus influenza* was grown on chocolate agar overnight and directly added to H/S, since it was not viable in PBS for 2 h. Stock solutions of sphingolipids in 5 mM in 7.5% n-octyl-β-D-glucopyranoside (OGP) or in dimethyl sulfoxide (DMSO) were prepared by sonication and warming to 40°C. SLs were added to the bacteria (0.025–200 µM in 0.075% OGP or 1% DMSO). Bacteria were incubated with SLs for 2 h at 37°C, and 200 µl of a bacteria solution plated on LB agar plates, incubated overnight at 37°C, and colonies counted.

**Interleukin-1β (IL-1β) and TNF-α measurements**

Lungs were removed 4 h after infection of mice or from uninfected mice, shock-frozen, lysed in 125 mM NaCl, 25 mM Tris–HCl (pH 7.4), 10 mM EDTA and 1% NP-40, and homogenized using a tip sonicator. Aliquots were analyzed for IL-1β and TNF-α by commercial ELISA kits (R&D).

**Statistics**

Data are means ± s.d. unless otherwise indicated. All data were tested with the David-Pearson-Stephens-test for normal distribution.

**The paper explained**

**Problem**

Cystic fibrosis (CF) is caused by mutations in the cystic fibrosis transmembrane conductance regulator and is the most common autosomal recessive disorder in western countries. At present, lung symptoms determine the quality of life and life expectancy of most CF patients, with patients displaying chronic inflammation and high susceptibility to lung infection with *Pseudomonas aeruginosa*, *Haemophilus influenzae*, *Burkholderia cepacia*, *Staphylococcus aureus*, and other bacteria. Approximately 80% of CF patients suffer from chronic *P. aeruginosa* pneumonia by the age of 25. Pulmonary *Pseudomonas aeruginosa* infections are also of major clinical importance in patients with chronic obstructive pulmonary disease, trauma, burn wounds, sepsis, or in patients requiring ventilation.

**Results**

Lung epithelial cells from human cystic fibrosis patients and cystic fibrosis mice display reduced sphingosine levels due to reduced acid ceramidase activity, which is reversed by acid ceramidase or sphingoid long chain base inhalation. Sphingoid long chain bases kill a broad spectrum of pathogenic bacteria at nanomolar to low micromolar concentrations, including *Pseudomonas aeruginosa*, *Acinetobacter baumannii*, *Haemophilus influenzae* and *Moraxella catarrhalis* and even *Burkholderia cepacia*. Inhalation of cystic fibrosis mice with acid ceramidase or sphingosine prevents and cures pulmonary *Pseudomonas aeruginosa* infection.

**Impact**

Tracheal and bronchial epithelial sphingosine acts as a natural antibacterial agent to prevent bacterial lung infection in healthy individuals. Sphingoid long chain bases show a broad anti-bacterial activity, and inhalation of sphingoid long chain bases may provide a novel therapeutic option to prevent or cure pulmonary bacterial infections.

All exact *P*-values are indicated in the figures and were calculated using a two-tailed distribution with a two sample equal or unequal variance Student's *t*-test (when the same sample or strains of mice were compared in different conditions or when different samples or strains of mice where compared in the same conditions, respectively). 0.05 was considered statistically significant.

**Supplementary information** for this article is available online: http://embomolmed.embopress.org

**Acknowledgements**

This study was supported by grant number 1105-69.11/2010 from the German-Israeli-Foundation to A.H.F. and E.G., DFG grant 335/16-2 to E.G., BMBF-grant 0315827B to B.T. and E.G., and NIH grant DK25809 to E.S. A.H. Futerman is the Joseph Meyerhoff Professor of Biochemistry at the Weizmann Institute of Science.

**Author contributions**

STT performed the sphingosine immunofluorescence experiments on human tissues and on some CF mouse tissues, KAB and HG performed confocal microcopy studies, TJ tested the effects of LCBs on *P. aeruginosa* growth *in vitro*, SL contributed human samples, BT contributed mice and *P. aeruginosa* strains, ES purified and provided human recombinant acid ceramidase, and KAB contributed to the *in situ* kinase assays. LJ and BK performed MS. MJE and ABL contributed to $EC_{50}$ measurements, JS characterized bacteria. YPJ and EG performed all other experiments. YPJ, EG, MJE, and AHF

     

designed the study and wrote the manuscript. All authors commented on the manuscript.

## Conflict of interest

The authors declare that they have no conflict of interest.

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
