## [Review Process File · EMBO Molecular Medicine]

Sphingoid long chain bases prevent lung infection by *Pseudomonas aeruginosa*

Yael Pewzner-Jung, Shaghayegh Tavakoli Tabazavareh, Heike Grassmé, Katrin Anne Becker, Lukasz Japtok, Jörg Steinmann, Tammar Joseph, Stephan Lang, Burkhard Tuemmler, Edward H. Schuchman, Alex B. Lentsch, Burkhard Kleuser, Michael Edwards, Anthony H. Futerman and Erich Gulbins

Corresponding author: Erich Gulbins, University of Duisburg-Essen and Yael Pewzner-Jung, Weizmann Institute of Science

Review timeline:	Submission date:	17 March 2014
	Editorial Decision:	23 March 2014
	Appeal:	24 March 2014
	Author Correspondence:	24 March 2014
	Additional Editorial Correspondence:	27 March 2014
	Editorial Decision:	09 April 2014
	Revision received:	17 June 2014
	Editorial Decision:	02 July 2014
	Accepted	11 July 2014

Transaction Report:

Editor: Céline Carret

1st Editorial Decision	23 March 2014
---------------

Thank you for the submission of your manuscript "Sphingoid long chain bases prevent lung infection by *Pseudomonas aeruginosa*".

I have now had the opportunity to carefully evaluate your paper and the related literature and I have also discussed it with my colleagues. I am afraid that we concluded that the manuscript is not well suited for publication in EMBO Molecular Medicine and have therefore decided not to proceed with peer review.

We appreciate that your data report reduced levels of sphingosine in the lung epithelial cells of cystic fibrosis patients and mice. This observation correlates with reduced acid ceramidase activity and therapeutic inhalation of either sphingosine analog or acid ceramidase restores levels and reduce *P. aeruginosa* lung infection in mice. Unfortunately, besides that the data remain at a descriptive level, sphingosine was already suggested to have antimicrobial properties and is well established as a signalling molecule involved in several cellular processes including inflammation. Furthermore, components of the sphingosine pathway have been implicated in airway hyper responsiveness and lung infection and new drugs have been developed to specifically target sphingomyelinases for the treatment of COPD, cystic fibrosis, asthma and acute lung injury. As such, we feel that the study

does not provide the sort of conceptual advance and mechanistic insights we expect in an EMBO Molecular Medicine article.

I am sorry that I could not bring better news.

Appeal

24 March 2014

Thank you for your email received yesterday about EMM-2014-04075. We understand that EMM does not send all papers out for peer review, but in this case, we feel the reasons that you outlined in your letter are far from being justified, and we would like to respectfully request that you reconsider your position.

We are more than familiar with the other papers that have been published on the role of the 'sphingosine pathway' on both infection and also on lung biology, since a number of them were written by authors of the current ms! However, we feel that your statement that 'components of the sphingosine pathway have been implicated in airway hyper responsiveness and lung infection' show both lack of understanding of the pathway, and also a lack of respect for lipids -- it would be similar to perhaps saying that 'protein kinases have been shown to be involved in signaling pathways and therefore any new details about the pathway, or about its role in pathology' is unworthy of a journal like EMM. We strongly feel that the 'devil is in the details', and in our study we have shown the remarkable and novel observation that sphingosine is an anti-bacterial agent in NORMAL lung epithelia; moreover, the idea of making novel long chain bases analogs for use as an anti-bacterial agent in lung infection is totally novel.

We would therefore like to request that this ms is sent out for peer review -- if our colleagues in the sphingolipid field feel the same as you and your editorial colleagues, then we will naturally accept their decision -- however, we feel it is grossly unfair to decide to decline our paper based on an uninformed view of the subtleties of the role of sphingolipids in physiology and pathophysiology.

Author Correspondence

24 March 2014

Please find below a letter to your editorial board regarding a manuscript, which we recently submitted to EMM.

We demonstrate the completely novel finding that sphingosine functions as one of the most important anti-bacterial agents in normal airways and demonstrate that this line of defense is almost absent in cystic fibrosis. The sensitivity of cystic fibrosis mice to infection is corrected by inhalation of sphingosine or acid ceramidase generating sphingosine in cystic fibrosis airways. This data identifies a completely novel defense mechanism in healthy lungs, its disruption in cystic fibrosis and the mechanism why this defense is disrupted in cystic fibrosis. Sphingosine is biologically not similar to sphingosine-1-phosphate at all. Its biological function is very poorly characterised, but the few data we know do not show any similarity with sphingosine-1-phosphate. It has never been implied in the defense of airways against bacteria and never been implicated to play a role in cystic fibrosis. Sphingosine has also never been implied in asthma, COPD or cystic fibrosis. Our data are the 1st to show a role in lung functions at all and they also provide the 1st insight into a role of sphingosine in lung diseases.

We agree that a finding on sphingosine-1-phosphate would not be novel. However, this does not apply to sphingosine at all.

We apologise, if we failed to indicate this point more clearly in the manuscript and we would be certainly happy to correct this. However, we very strongly feel that the science of this manuscript is highly novel, mechanistic and clinically relevant.

Thus, we kindly ask to reconsider your decision.

Additional Editorial Correspondence

27 March 2014

This e-mail is to let you know that following internal discussions involving our Chief Editor, in light of your comments and our phone call conversation, we decided to give your manuscript a chance and we will send it out for review.

I will contact you again as soon as the reports will be in.

2nd Editorial Decision

09 April 2014

Thank you for the submission of your manuscript to EMBO Molecular Medicine. We have now heard back from the three referees whom we asked to evaluate your manuscript. Although the referees find the study to be of interest, they also raise a number of concerns that must be addressed in the next final revised version of your manuscript.

As you will see, all three referees are rather supportive of publication. However, they highlight some technical issues (experiments chosen, additional controls, better stats) and require additional explanations/clarifications here and there. Referee 3 suggests adding data on the host response to the administration of sphingoid long chain bases and we agree that it would nicely complement the study.

Should you be able to address these criticisms in full, we would be willing to consider a revised manuscript. However, please note that that it is our journal's policy to allow only a single round of revision, and that acceptance or rejection of the manuscript will therefore depend on the completeness of your response and the satisfaction of the referees with it.

I look forward to receiving your revised manuscript.

***** Reviewer's comments *****

Referee #1 (Comments on Novelty/Model System):

This is an interesting and well written paper showing that long chain sphingoid bases can prevent infections by *P. aeruginosa* in a cystic fibrosis mouse model. However, there are several issues that need to be addressed.

1. Measurements of sphingosine and ceramide were by old fashioned and relatively non-specific enzymatic methods. Since this is an important point of the paper and species of ceramide and sphingoid bases might have important functions, it is important to measure their levels as in Figure 2 by quantitative mass spectrometry.
2. It is not clear why the results with the CerS2 knockout mice are important or relevant for this paper. Moreover, the ceramide levels described in this paper are not consistent with previous reports from the same authors.

Referee #1 (Remarks):

This is an interesting and well written paper showing that long chain sphingoid bases can prevent infections by *P. aeruginosa* in a cystic fibrosis mouse model. However, there are several issues that

need to be addressed.

1. Measurements of sphingosine and ceramide were by old fashioned and relatively non-specific enzymatic methods. Since this is an important point of the paper and species of ceramide and sphingoid bases might have important functions, it is important to measure their levels as in Figure 2 by quantitative mass spectrometry.

2. It is not clear why the results with the CerS2 knockout mice are important or relevant for this paper. Moreover, the ceramide levels described in this paper are not consistent with previous reports from the same authors.

3. FTY720 might have other actions than sphingosine as the data in Table 1 suggests that it is 6-fold less effective than sphingoid bases in vitro whereas in Figure 3 it has similar potency. The effects of FTY720 on lymphocyte trafficking might also affect recruitment of inflammatory cells to sites of infection.

4. The results shown in supplement Figure 3 are crucial and should be shown in the manuscript as they might explain the mechanism by which sphingosine prevents pseudomona infection. However, the concentration of sphingosine used in this figure is 50 times greater than the EC50 determined in Table 1. Therefore, similar experiments should be carried out with concentrations around the EC50. In this regard, it is also important to show that concentrations of sphingoid bases that prevent pseudomona growth in vitro do not cause damage to epithelial cells.

Minor point

The description of the methods for the enzymatic measurements of sphingosine and ceramide do not seem correct. There is no Mg in the buffer and sodium acetate cannot be used as a buffer at pH 7.4.

There is no discussion of the results with strains PA14 and ATCC853 in Figure 3. Some explanation for why this experiment is important should be provided.

The description of the enzymatic assay for sphingosine should be changed from "SK assay" to "enzymatic assay of sphingosine". In any case, these old fashioned methods should be replaced with more modern techniques.

Referee #2 (Remarks):

Summary:

Sphingosine (SPH) has been identified as an anti-bacterial agent (Arikawa et al. J. Invest. Dermatol. 2002 119:433-439). In this research, the authors demonstrate the role of SPH in modulating susceptibility to lung infection by *Pseudomonas aeruginosa*. By immunofluorescence staining and biochemical measurement, they determine the SPH level in nasal and lung tissue derived from humans and mice both from normal and CFTR-deficient samples. They find SPH levels were highly decreased in CF patients and cystic fibrosis mouse models and could be restored by in vivo treatment with acid ceramidase, which is responsible for the production of SPH from ceramide. Furthermore, they find that inhalation of SPH or acid ceramidase protected susceptible mice from *Pseudomonas aeruginosa* infection and even could cure already existing infections. These results suggest that sphingosine can prevent *P.aeruginosa* infection and might be used as a novel treatment option for this type of infection.

Major comments:

1. The direct antibacterial effect of the different LCBs is remarkable. Does this also holds true for other gram-negative airway infecting agents associated with CF or COPD, e.g. *Haemophilus influenzae*, *Burkholderia cepacia* or *Moraxella catarrhalis* ? It would also be nice to know if bacteria incubated +/- ceramide in the presence of acid ceramidase (either active or heat inactivated) will also be killed.

2. Fig. 2: Application of ceramide in low pH buffer increases SPH levels. Could it be that the low pH also leads to chemical hydrolysis of ceramide instead of the assumed enzymatic hydrolysis by acid ceramidase? It would be good to include such an experimental control (incubation of ceramide at different pH values in the presence/absence of acid ceramidase) to rule out a chemical hydrolysis of ceramide under these conditions).

3. The statistics need to be checked. Given the large standard deviations as indicated (e.g. in Fig. 3A and 3B) it is unclear to me, if the values (n=4) have a normal distribution and allow the use of the students t-test.

Minor comments:

1. Page 3: "Among these are the sphingoid long chain base (LCB)," should be ".....bases"
2. Page 3: "However, the increasing rate of *P. aeruginosa* resistance to many antibiotics renders essential the need to find alternative therapies..."
this awkward sentence should be changed to something like:
"However, the increasing rate of *P. aeruginosa* resistance to many antibiotics demands/necessitates alternative therapies..."
3. Page 4: "...SPH and ceramide kinase assays on extracts from freshly isolated tracheal..." maybe better "...SPH and ceramide kinase assays using extracts from freshly isolated tracheal..."
4. Page 5: "These results demonstrate that SPH is present on the surface of WT epithelial cells while almost completely absent on the surface of CF epithelia" should be "These results demonstrated that SPH is present on the surface of WT epithelial cells while almost completely absent on the surface of CF epithelia"
5. It would be useful to check the tense throughout the whole manuscript; results should be presented in past tense!
6. Page 6: "...providing structural information that could of use in development of novel LCB-based drugs to treat lung infection" should be "...providing structural information that could be used in development of novel LCB-based drugs to treat lung infection".
7. Page 6: "...anti-bactericidal effect of lung and trachea epithelial" should be "...anti-bacterial effect of lung and trachea epithelial" or "...bactericidal effect of lung and trachea epithelial"
8. Page 7: "...are highly efficacious in killing bacteria but will not affect the host",
maybe it is better "...are highly efficacious in killing bacteria without causing effects on the host"

Referee #3 (Remarks):

The manuscript by Pewzner-Jung et. al. describes the antimicrobial ability of sphingoid long base chains to kill *P. aeruginosa* and proposes that reduced expression of sphingosine in the airways of Cystic Fibrosis patients and CF mice. Importantly, their data demonstrate that addition of synthetic sphingoid long base chains (LBCs) can improve bacterial clearance from the airways of CF mice, suggesting that they could be used as a potential treatment in these patients. Additionally, data describing how structure and charge influences killing will be essential in the design and optimization of synthetic structures. A few additions would strengthen the manuscript:

- 1- The manuscript was lacking data describing the response of the host, if any, to the synthetic LBCs. It would be useful to document the inflammatory state of the lungs following administration of the LBCs.
- 2- Markers of inflammation such as cytokine expression and images of lung histology from the infected and treated mice would be helpful.

1st Revision - authors' response

17 June 2014

Referee #1 (Comments on Novelty/Model System):

*This is an interesting and well written paper showing that long chain sphingoid bases can prevent infections by *P. aeruginosa* in a cystic fibrosis mouse model. However, there are several issues that need to be addressed.*

1. Measurements of sphingosine and ceramide were by old fashioned and relatively non-specific enzymatic methods. Since this is an important point of the paper and species of ceramide and sphingoid bases might have important functions, it is important to measure their levels as in Figure 2 by quantitative mass spectrometry.

Response:

We have now performed MS and show quantitative analysis of sphingosine and ceramide species in isolated epithelial cells. The data confirm the previous biochemical and histological data and also demonstrate that C16, C18, C22, C24 and C24:1 ceramide-species accumulate in tracheal epithelial cells of cystic fibrosis mice.

2. It is not clear why the results with the CerS2 knockout mice are important or relevant for this paper. Moreover, the ceramide levels described in this paper are not consistent with previous reports from the same authors.

Response:

We used the CerS2 null mouse as a 2nd genetic model which displays a marked reduction of sphingosine in its airways, but is genetically completely independent of cystic fibrosis mice, in order to show that sphingosine is a major player in the defense against bacterial lung infections.

The data that was published by some of the authors showed total ceramide levels in whole lung (where more than 90% of the tissue are alveoli). In the present manuscript we document the levels of ceramide specifically in the bronchi, and show an increase in the epithelial cells of this particular regions of the lung. This data demonstrates a specific increase of ceramide in a small fraction of the lung in CerS2 null mice, which might be obscured by changes of ceramide in the aveoli of CerS2 null mice, since CerS2 deficiency affects C24-SLs levels in all cells in the lung with a variable degree of compensatory increase of C16 ceramide in different cells. This results in a complex mixture of changes in the whole lung compared to changes in a certain cell type. In contrast, Cfr is only expressed in bronchial epithelial cells and alveolar macrophages and the absence of Cfr does not affect ceramide in other cells of the lung.

We now indicate that we determined ceramide specifically in bronchial and epithelial cells in Cers2 null mice.

3. FTY720 might have other actions than sphingosine as the data in Table 1 suggests that it is 6-fold less effective than sphingoid bases in vitro whereas in Figure 3 it has similar potency. The effects of FTY720 on lymphocyte trafficking might also affect recruitment of inflammatory cells to sites of infection.

Response:

We now show that there is no cytokine increase in the lung and no recruitment of neutrophils or monocytes and lymphocytes (anti-Gr-1 staining and Hemalaun – see revised Fig. 3C, Supplementary Fig. 4, 5 and 6) into the lung after inhalation of FTY720 prior or after infection with *P. aeruginosa*, excluding the possibility that neutrophils or any other innate immune cells or early-recruited lymphocytes mediate the anti-bacterial effect of the inhalation of FTY-720.

4. The results shown in supplement Figure 3 are crucial and should be shown in the manuscript as they might explain the mechanism by which sphingosine prevents pseudomona infection. However, the concentration of sphingosine used in this figure is 50 times greater than the EC50 determined in Table 1. Therefore, similar experiments should be carried out with concentrations around the EC50. In this regard, it is also important to show that concentrations of sphingoid bases that prevent pseudomonas growth in vitro do not cause damage to epithelial cells.

Response:

Based on this comment, we went back and evaluated the use of Sytox Green as a marker for bacterial viability and permeabilization. Apparently, it appears that Sytox Green is not as good marker as we previously thought (Leberon et al, Applied Environmental Biology 64, 2967). Sytox Green underestimates the fraction of dead cells. Based on this, we performed a new set of experiments using Sytox Green with doses as low as 1 micromolar of sphingosine and examined the number of green cells by FACS analysis. Although we detected significant levels of fluorescence in the bacteria upon 1 micromolar sphingosine treatment, the EC50 was higher than the EC50 obtained with assays based on growing bacteria on agar plates (the gold standard for assaying bacterial death and growth inhibition), consistent with the published demonstration that Sytox Green underestimates death. This is explained by the fact that Sytox Green does not stain cells that are dead, but still have an intact membrane, or bacteria in which the DNA has been completely degraded. Because of these inherent limitations of the use of Sytox Green, we feel that our previous conclusion that the mechanism of cell death definitively involves membrane permeabilization cannot be fully substantiated by our current data and therefore we have decided to remove this data from the current manuscript. Ongoing studies are underway to determine the exact mechanism of bacterial death.

We have left a sentence in the discussion that the mechanism of bacterial death might be either by direct or indirect effects of sphingosine.

Minor point

The description of the methods for the enzymatic measurements of sphingosine and ceramide do not seem correct. There is no Mg in the buffer and sodium acetate cannot be used as a buffer at pH 7.4.

Response:

We titrated the sodium acetate to pH 7.4 using NaOH. We now replaced the word 'buffer' by the word 'solution'.

The airway liquid contains Mg²⁺ at 1.9 mmol/liter (Bacconnais, S., R. Tirouvanziam, J.-M. Zahm, S. de Bentzmann, B. Péault, G. Balossier, and E. Puchelle. 1999. Ion composition and rheology of airway liquid from cystic fibrosis fetal tracheal xenografts. *Am. J. Respir. Cell Mol. Biol.* 20:605–611). Since we added a very small volume in all *in situ* kinase assays, we left the Mg²⁺-concentration high enough to perform the *in situ* kinase assays. In an attempt to leave the airway surface liquid unaltered as much as possible, we did not add any further Mg²⁺.

However, we also performed controls in which we added Magnesium phosphate, but it did not make a difference. We now show the kinase data including MgPO₄ in the supplementary information (S-Fig. 7).

The kinase buffer used to determine sphingosine in lysates from isolated epithelial cells contained Mg²⁺ (50 mM HEPES (pH 7.4), 250 mM NaCl, 30 mM MgCl₂, 1 mM ATP and 10 μCi [³²P]γATP).

There is no discussion of the results with strains PA14 and ATCC853 in Figure 3. Some explanation for why this experiment is important should be provided.

Response:

The strains were used to prove that we do not have strain-specific results. This is now indicated in the text.

The description of the enzymatic assay for sphingosine should be changed from "SK assay" to "enzymatic assay of sphingosine". In any case, these old fashioned methods should be replaced with more modern techniques.

Response:

We changed the text according to the suggestion of the referee. We now also analyzed sphingosine and ceramide species with mass spectrometry and confirm the sphingosine and ceramide measurements using enzymatic assays and fluorescence microscopy.

Referee #2 (Remarks):

Summary

Sphingosine (SPH) has been identified as an anti-bacterial agent (Arikawa et al. J. Invest. Dermatol. 2002 119:433-439). In this research, the authors demonstrate the role of SPH in modulating susceptibility to lung infection by Pseudomonas aeruginosa. By immunofluorescence staining and biochemical measurement, they determine the SPH level in nasal and lung tissue derived from humans and mice both from normal and CFTR-deficient samples. They find SPH levels were highly decreased in CF patients and cystic fibrosis mouse models and could be restored by in vivo treatment with acid ceramidase, which is responsible for the production of SPH from ceramide. Furthermore, they find that inhalation of SPH or acid ceramidase protected susceptible mice from Pseudomonas aeruginosa infection and even could cure already existing infections. These results suggest that sphingosine can prevent P. aeruginosa infection and might be used as a novel treatment option for this type of infection.

Major comments:

1. The direct antibacterial effect of the different LCBs is remarkable. Does this also holds true for other gram-negative airway infecting agents associated with CF or COPD, e.g. Haemophilus influenzae, Burkholderia cepacia or Moraxella catarrhalis ? It would also be nice to know if bacteria incubated +/- ceramide in the presence of acid ceramidase (either active or heat inactivated) will also be killed.

Response:

We now tested the effects of sphingosine on *Acinetobacter baumannii*, *Haemophilus influenzae*, *Burkholderia cepacia* and *Moraxella catarrhalis*. The studies demonstrate that *Acinetobacter baumannii* and *Moraxella catarrhalis* are very sensitive to sphingosine and are killed at nanomolar doses, while *Haemophilus influenzae* displays an intermediate sensitivity with killing at low micromolar concentrations. In contrast, *Burkholderia cepacia* is relatively resistant with an EC₅₀ of 45 micromolar. *Burkholderia* has a specialized membrane and capsule and potential reasons for its resistance are discussed.

Heat inactivated acid ceramidase was without any effect.

Incubation of the bacteria with acid ceramidase in the presence of ceramide resulted in sphingosine release and killing of the bacteria.

2. Fig. 2: Application of ceramide in low pH buffer increases SPH levels. Could it be that the low pH also leads to chemical hydrolysis of ceramide instead of the assumed enzymatic hydrolysis by acid ceramidase? It would be good to include such an experimental control (incubation of ceramide at different pH values in the presence/absence of acid ceramidase) to rule out a chemical hydrolysis of ceramide under these conditions).

Response:

To test whether an acid-mediated hydrolysis of ceramide contributes to the consumption of ceramide and the generation of sphingosine, we performed the experiment in the presence of the acid ceramidase inhibitors carmofur or oleoyethanolamine. The inhibitors completely prevented the generation of sphingosine indicating that the generation of sphingosine is mediated by acid ceramidase. This is now shown in Fig. 2D.

3. The statistics need to be checked. Given the large standard deviations as indicated (e.g. in Fig. 3A and 3B) it is unclear to me, if the values ($n=4$) have a normal distribution and allow the use of the students *t*-test.

Response:

We now tested all data with David-Pearson-Stephens-test for normal distribution. The results show normal distribution permitting us to perform *t*-test for single comparisons or ANOVA for multiple comparisons.

Minor comments:

1. Page 3: "Among these are the sphingoid long chain base (LCB)," should be ".....bases"

2. Page 3: "However, the increasing rate of *P. aeruginosa* resistance to many antibiotics renders essential the need to find alternative therapies..."

this awkward sentence should be changed to something like:

"However, the increasing rate of *P. aeruginosa* resistance to many antibiotics demands/necessitates alternative therapies..."

3. Page 4: "...SPH and ceramide kinase assays on extracts from freshly isolated tracheal..." maybe better "...SPH and ceramide kinase assays using extracts from freshly isolated tracheal..."

4. Page 5: "These results demonstrate that SPH is present on the surface of WT epithelial cells while almost completely absent on the surface of CF epithelia" should be "These results demonstrated that SPH is present on the surface of WT epithelial cells while almost completely absent on the surface of CF epithelia"

5. It would be useful to check the tense throughout the whole manuscript; results should be presented in past tense!

6. Page 6: "...providing structural information that could of use in development of novel LCB-based drugs to treat lung infection" should be "...providing structural information that could be used in development of novel LCB-based drugs to treat lung infection".

7. Page 6: "...anti-bactericidal effect of lung and trachea epithelial" should be "...anti-bacterial effect of lung and trachea epithelial" or "...bactericidal effect of lung and trachea epithelial"

8. Page 7: "...are highly efficacious in killing bacteria but will not affect the host", maybe it is better "...are highly efficacious in killing bacteria without causing effects on the host"

Response:

We corrected in the text all the minor comments suggested above.

Referee #3 (Remarks):

The manuscript by Pewzner-Jung et. al. describes the antimicrobial ability of sphingoid long base chains to kill *P. aeruginosa* and proposes that reduced expression of sphingosine in the airways of Cystic Fibrosis patients and CF mice. Importantly, their data demonstrate that addition of synthetic

sphingoid long base chains (LBCs) can improve bacterial clearance from the airways of CF mice, suggesting that they could be used as a potential treatment in these patients. Additionally, data describing how structure and charge influences killing will be essential in the design and optimization of synthetic structures. A few additions would strengthen the manuscript:

- 1- *The manuscript was lacking data describing the response of the host, if any, to the synthetic LBCs. It would be useful to document the inflammatory state of the lungs following administration of the LBCs.*

Response:

We now performed histology analysis on lungs from mice that received sphingosine, acid ceramidase or FTY-720 inhalation. The lungs were investigated 4 hrs, 1 d and 14 days after inhalation. We performed hemalaun staining and immunostaining with anti-Gr-1 antibodies to detect influx of leukocytes into the lung after application of the long chain bases. In addition, we measured Interleukin-1 and TNF α in lung homogenates 4 hrs, 1 d and 14 days after inhalation of the long chain bases.

Collectively, the data show that the inhalation of sphingosine, acid ceramidase or FTY-720 has no pro-inflammatory effect on the lung.

This data are now shown in Fig. 3C and supplementary Fig. 4, 5 and 6 of the revised manuscript.

- 2- *Markers of inflammation such as cytokine expression and images of lung histology from the infected and treated mice would be helpful.*

Response:

We determined IL-1 and TNF α levels in lung homogenates 4 hrs after infection with *P. aeruginosa* strains 762 or ATCC 27853. Mice were inhaled prior or after infection with sphingosine, acid ceramidase or FTY-720. We also determined the cytokines in non-infected mice that were inhaled with the compounds or left untreated. Further, we performed hemalaun stainings and immunostainings of the lung prior and after infection \pm application of the long chain bases to detect influx of inflammatory cells, in particular neutrophils.

The data show massive inflammation in the lung after infection with *P. aeruginosa*. Cystic fibrosis mice showed some mild inflammation prior to the infection, but a very severe inflammation after infection, which is much more pronounced than in wild type mice consistent with the higher number of bacteria in the lungs of these mice. Inhalation of sphingosine, acid ceramidase or FTY-720 almost completely prevented the release of IL-1 and TNF α in lungs as well as the influx of leukocytes into the lungs consistent with the bactericidal effect of sphingosine already in the upper airways.

3rd Editorial Decision

02 July 2014

Thank you for the submission of your revised manuscript to EMBO Molecular Medicine. We have now received the enclosed reports from the referees that were asked to re-assess it. As you will see the reviewers are now supportive and I am pleased to inform you that we will be able to accept your manuscript pending final editorial amendments.

Please submit your revised manuscript within two weeks.

I look forward to reading a new revised version of your manuscript soon.

***** Reviewer's comments *****

Referee #1 (Remarks):

The authors adequately addressed all previous concerns.

Referee #2 (Remarks):

The authors have, in their revised version and in their response letter, appropriately addressed all my previous concerns.